# Clinical Evaluation of Functional Lumbar Segmental Instability: Reliability, Validity, and Subclassification of Manual Tests—A Scoping Review

**DOI:** 10.3390/jfmk10040400

**Published:** 2025-10-15

**Authors:** Ioannis Tsartsapakis, Aglaia Zafeiroudi, Gerasimos V. Grivas

**Affiliations:** 1Department of Physical Education and Sport Science, Aristotle University of Thessaloniki, 62100 Serres, Greece; 2Department Physical Education and Sport Science, University of Thessaly, 42100 Trikala, Greece; azafeiroudi@uth.gr; 3Physical Education and Sports, Division of Humanities and Political Sciences, Hellenic Naval Academy, 18539 Piraeus, Greece; grivasger@hotmail.com

**Keywords:** segmental motor control, manual spine examination, neuromuscular coordination, functional spinal instability, musculoskeletal screening tools, rehabilitation stratification

## Abstract

**Background**: Functional lumbar segmental instability (FLSI) is a clinically significant subtype of nonspecific low back pain, characterized by impaired motor control during mid-range spinal motion. Despite its prevalence, diagnostic approaches remain fragmented, and no single clinical test reliably captures its complexity. This scoping review aims to synthesize current evidence on the reliability, validity, subclassification, and predictive value of manual tests used in the evaluation of FLSI, and to identify conceptual and methodological gaps in the literature. **Methods**: A structured search was conducted across five databases (PubMed, Scopus, Web of Science, CINAHL, Embase) between May and August 2025. Twenty-four empirical studies and eleven foundational conceptual sources were included. Data were charted into five thematic domains: conceptual frameworks, diagnostic accuracy, reliability, subclassification models, and predictive value. Methodological appraisal was performed using QUADAS and QAREL tools. **Results**: The Passive Lumbar Extension Test (PLET) demonstrated the most consistent reliability and clinical utility. The Prone Instability Test (PIT) and Posterior Shear Test (PST) showed variable performance depending on protocol standardization. Subclassification models distinguishing functional, structural, and combined instability achieved high inter-rater agreement. Screening tools for sub-threshold lumbar instability (STLI) showed preliminary feasibility. Predictive validity of manual tests for rehabilitation outcomes was inconsistent, suggesting the need for multivariate models. **Conclusions**: Manual tests can support the clinical evaluation of FLSI when interpreted within structured diagnostic frameworks. Subclassification models and composite test batteries enhance diagnostic precision, but standardization and longitudinal validation remain necessary. Future research should prioritize protocol harmonization, integration of sensor-based technologies, and stratified outcome studies to guide individualized rehabilitation planning.

## 1. Introduction

Lumbar segmental instability (LSI) refers to a biomechanical condition characterized by excessive intervertebral motion beyond physiological limits under load, frequently associated with pain, functional impairment, and recurrent episodes of low back pain (LBP) [1,2]. Clinically, LSI is divided into two subtypes: structural instability, which involves excessive end-range motion due to compromised passive stabilizers, and functional instability, which reflects impaired motor control during mid-range movement [3,4].

To reduce conceptual overlap, this review adopts the term functional lumbar segmental instability (FLSI) to describe mid-range motor control deficits that are not detectable through imaging but manifest during dynamic tasks. While both subtypes originate from biomechanical dysfunction, their diagnostic approaches diverge significantly [3,5]. Structural instability is typically confirmed through radiographic indicators such as sagittal translation or angular displacement [6]. In contrast, FLSI requires clinical evaluation based on movement observation and manual testing, as radiographic methods are insufficient for detecting mid-range dysfunction [7,8].

Patients with FLSI often report nonspecific LBP, aberrant movement patterns, and reduced control during dynamic tasks [4,9,10]. The reliability and validity of manual clinical tests vary across studies, and no universally accepted diagnostic standard currently exists [11,12,13,14].

Several manual tests have been proposed to assess FLSI, including the Prone Instability Test (PIT), Passive Lumbar Extension Test (PLET), Aberrant Movement Patterns (AMP), and motor control assessments such as the Deep Muscle Contraction (DMC) scale and the Clinical Test of Thoracolumbar Dissociation (CTTD) [5,10]. Among these, the PLET has shown strong diagnostic accuracy for structural instability, though its relevance to functional deficits remains under investigation [8,12]. These tests have also been evaluated for their ability to predict rehabilitation outcomes, particularly in relation to lumbar stabilization exercises (LSE), although findings remain inconclusive [10,15,16].

Recent studies have introduced subclassification models that distinguish between functional, structural, and combined instability, with promising inter-rater reliability [3,17]. Screening tools for sub-threshold lumbar instability (STLI) have also been developed to identify early-stage dysfunction before radiographic signs appear [18,19]. These approaches support the development of integrative diagnostic frameworks that combine clinical observation, motor control testing, and functional assessment [14].

Additional research has explored symptom patterns associated with lumbar instability in athletic populations [20] and identified preliminary prognostic indicators for rehabilitation outcomes in chronic LBP [21].

Despite the growing body of literature, there is a lack of integrative reviews that critically synthesize manual tests for FLSI in relation to reliability, validity, subclassification, and predictive value. Moreover, the conceptual foundations of these tests are rarely linked to biomechanical models of spinal stability, such as Panjabi’s framework [22], nor are they consistently evaluated within stratified clinical contexts. This gap limits the development of standardized protocols and impairs clinical decision-making.

To address these limitations, this review is guided by three core questions:

(1) What is the current evidence regarding the reliability and validity of manual tests for FLSI?

(2) How can these tests be organized into clinically meaningful subclassification domains?

(3) What are the implications of current evidence for clinical decision-making and future research?

This narrative review includes 24 peer-reviewed studies published between 2000 and 2025. It aims to synthesize current evidence on the clinical evaluation of FLSI, with specific attention to the reliability, validity, and subclassification of manual tests used in outpatient settings. It also examines their predictive value for rehabilitation outcomes and discusses their role in clinical decision-making. In addition to empirical studies, foundational sources were referenced to support the conceptual and biomechanical framework. The review synthesizes biomechanical, neuromuscular, and clinical perspectives to support evidence-informed physical therapy practice in musculoskeletal rehabilitation.

## 2. Materials and Methods

### 2.1. Review Design

This study was designed as a descriptive narrative review, aiming to synthesize and critically appraise the existing literature on the clinical assessment of FLSI. The emphasis of this analysis was on the reliability, validity, and subclassification of manual tests. Systematic reviews are distinguished by their adherence to stringent inclusion criteria and the implementation of meta-analytic procedures. In contrast, narrative reviews adopt a more expansive approach, facilitating conceptual integration and thematic organization of findings across diverse study designs. This approach is particularly well-suited to conditions such as FLSI, where diagnostic ambiguity and variability in assessment methods preclude quantitative synthesis.

To address concerns regarding methodological clarity, we acknowledge that the present review incorporates semi-systematic features (e.g., structured search strategy, eligibility criteria, thematic coding), and may be more accurately described as a scoping narrative review. This hybrid format was selected to balance conceptual breadth with methodological transparency, in line with recent recommendations for reviews of complex clinical phenomena [23].

The thematic domains were determined a priori based on clinical relevance and recurring constructs identified in the literature. The review was structured into five domains: (a) conceptual definitions of FLSI, (b) diagnostic limitations of radiographic methods, (c) reliability and validity of manual clinical tests, (d) subclassification models, and (e) predictive value for rehabilitation outcomes.

### 2.2. Literature Search Strategy

The literature search was conducted systematically across five electronic databases: PubMed, Scopus, Web of Science, CINAHL, and Embase. The search was conducted between May and August 2025 and executed independently by three reviewers. The keyword combinations employed encompassed both free-text terms and controlled vocabulary (e.g., MeSH), tailored to the indexing systems of each database.

The search terms were expanded to include additional clinically relevant constructs such as ‘Clinical Test of Thoracolumbar Dissociation’ and ‘Deep Muscle Contraction scale,’ which are discussed in the Results section. These additions ensured alignment between the search strategy and the thematic scope of the review. These terms were incorporated during the second phase of search refinement, following preliminary thematic coding and identification of emerging constructs in the literature.

The final search terms included: “functional lumbar segmental instability,” “lumbar instability and clinical tests,” “motor control impairment and low back pain,” “lumbar stabilization exercises and prediction,” “inter-rater reliability and lumbar spine,” “subclassification and lumbar instability,” “Clinical Test of Thoracolumbar Dissociation,” and “Deep Muscle Contraction scale.” Boolean operators and truncation symbols were employed to optimize sensitivity and capture relevant variations. To restrict the results to English or German language articles, a filter was applied to limit the publication range. Language filters restricted results to English or German. A manual review of reference lists from included articles and relevant reviews was conducted to identify supplementary sources. Article selection was performed independently by two reviewers, with discrepancies resolved by consensus involving a third reviewer. This multi-step process ensured comprehensive coverage of the literature relevant to the clinical evaluation of FLSI. A variety of foundational theoretical sources were referenced separately to support the conceptual framework and biomechanical definitions relevant to FLSI. These sources include White [2].

A total of 24 peer-reviewed studies were included in the final synthesis. In addition, 11 foundational theoretical sources were referenced to support the conceptual framework and biomechanical definitions relevant to FLSI. These sources were selected based on citation frequency, relevance to spinal stability models, and inclusion in prior reviews.

It should be noted that the 24 studies included in the final synthesis were selected through a structured eligibility screening process. In addition to these, a set of foundational theoretical sources was referenced to support the conceptual framework and interpretive synthesis. These sources were not subject to eligibility screening, as they did not meet the criteria for empirical inclusion, but were selected based on citation frequency, relevance to spinal stability models, and their role in prior conceptual reviews. Table 1 summarizes these referenced sources, which serve to contextualize the biomechanical and clinical constructs of FLSI.

Although this review does not follow a systematic protocol, a simplified flow diagram was constructed to illustrate the study selection process and enhance methodological transparency (Figure 1).

A structured database search yielded 327 records. After removal of duplicates and initial screening, 48 full-text articles were assessed for eligibility. Exclusion criteria included studies focused exclusively on surgical populations, lacking empirical data on manual test performance, or addressing structural instability without reference to motor control dysfunction. Editorials, conference abstracts, and expert opinions lacking methodological transparency were excluded.

Figure 1 summarizes the number of records identified, screened, assessed for eligibility, and included in the final synthesis.

### 2.3. Eligibility Criteria

The inclusion of studies in this review was contingent upon their alignment with predetermined criteria, encompassing criteria such as publication date, linguistic characteristics, population under study, the focal point of the study, the outcomes reported, and the design of the study. To be considered, studies had to be published between 1 January 2000, and 28 August 2025. They were required to be written in English or German and appear in peer-reviewed journals. The study population included adults aged 18 years or older who suffered from chronic or recurrent nonspecific LBP, irrespective of the presence of radiographic evidence of lumbar instability. In order to align with the scope of this review, studies were required to investigate clinical tests related to FLSI, motor control impairment, or segmental motion dysfunction. The eligible outcomes encompassed a range of metrics, including diagnostic accuracy metrics (e.g., sensitivity, specificity, likelihood ratios), reliability indices (e.g., kappa coefficients), predictive value for rehabilitation outcomes, and subclassification of instability. The study designs that were deemed acceptable encompassed original research, including cross-sectional, cohort, and diagnostic accuracy studies, as well as systematic or narrative reviews that clearly reported inclusion criteria and selection process.

Exclusion criteria encompassed studies that exclusively focused on surgical populations or postoperative instability, investigated structural instability without regard for functional or motor control elements, lacked empirical data regarding clinical test performance, or were constrained to conference abstracts, editorials, or expert opinions devoid of methodological transparency.

The eligibility screening did not apply to foundational textbooks and conceptual models, which were instead referenced to contextualize definitions of segmental instability and to support the theoretical interpretation of clinical findings. Although the structured search was limited to empirical studies published between 2000 and 2025, foundational theoretical sources predating this range were included to support the conceptual framework and interpretive synthesis, in accordance with accepted practices for scoping reviews of complex clinical phenomena [23].

Foundational conceptual sources, including biomechanical models and classification frameworks, were referenced separately and were not subject to the eligibility criteria applied to empirical studies. Their inclusion served to support the theoretical interpretation of clinical findings and the development of the diagnostic framework.

These criteria ensured that the included studies directly informed the clinical evaluation framework for FLSI.

### 2.4. Data Extraction and Thematic Categorization

For each study that met the established criteria, the review team manually extracted the relevant data and organized it into structured tables. This process was undertaken to facilitate the subsequent thematic synthesis. The following variables were extracted: author(s), year of publication, study design, sample characteristics, clinical tests evaluated, reference standards (if applicable), and reported outcomes related to reliability, validity, or predictive capacity. After data extraction, studies were systematically categorized into five distinct thematic domains, based on their predominant analytical focus. The present categorization was developed inductively, that is, based on recurring analytical themes identified across the included studies.

The first domain included conceptual frameworks, articles defining or discussing the biomechanical and neuromuscular basis of FLSI. The second focused on diagnostic accuracy, studies reporting sensitivity, specificity, likelihood ratios, or diagnostic odds ratios of clinical tests against radiographic or intraoperative reference standards. The third addressed reliability, incorporating research evaluating inter- or intra-rater agreement using kappa coefficients, weighted concordance, or percentage agreement. The fourth included subclassification models, and studies proposing or validating the distinction between functional, structural, and combined instability, often supported by inter-rater reliability metrics. The fifth examined predictive value for rehabilitation, studies assessing whether baseline clinical test results were associated with outcomes following lumbar stabilization exercises or other conservative interventions.

This thematic organization enabled structured synthesis across heterogeneous methodologies and facilitated the identification of converging evidence, methodological limitations, and gaps in the literature.

### 2.5. Quality Appraisal Approach

Quality appraisal was used not to exclude studies, but to contextualize their contribution to the thematic synthesis. Due to the narrative character of this review, formal risk-of-bias scoring was not applied uniformly across all included studies. Nonetheless, methodological quality was systematically considered during data synthesis, particularly for studies reporting diagnostic accuracy and reliability metrics. For diagnostic studies, the Quality Assessment of Diagnostic Accuracy Studies (QUADAS) tool was used to assess patient selection, index test clarity, reference standard transparency, and timing. For reliability studies, the QAREL framework was applied to evaluate examiner blinding, test standardization, and statistical rigor.

The specific criteria encompassed patient selection procedures, clarity in the description of the index test, transparency regarding the reference standard, and the timing of assessments [24]. The Quality Appraisal of Reliability Studies (QAREL) framework was consulted to assess reliability studies for studies of examiner blinding, standardization of testing procedures, and statistical analysis appropriateness [25]. Studies with incomplete reporting, unclear methodology or insufficient sample size were interpreted with caution and flagged during thematic synthesis. They were not excluded solely based on quality. Appraisal outcomes were documented during synthesis to inform interpretive weighting.

Table 1 presents a summary of the quality appraisal outcomes for key studies, including the type of evaluation tool used, criteria met, and interpretive weighting applied during synthesis.

**Table 1 jfmk-10-00400-t001:** Summary of Quality Appraisal Using QUADAS and QAREL Frameworks.

Study	Type	Tool Used	Criteria Met (Out of 7)	Interpretive Weight
Rabin et al. [26]	Reliability	QAREL	6	High interpretive weight
Alyazedi et al. [3]	Diagnostic	QUADAS	6	High interpretive weight
Oliveira et al. [10]	Diagnostic	QUADAS	4	Moderate—used with caution
Dankaerts et al. [27]	Reliability	QAREL	5	Moderate—supports subgrouping
Fritz et al. [28]	Diagnostic	QUADAS	3	Low—illustrates variability
Hodges & Richardson [29]	Conceptual	—	—	Qualitative appraisal only

Note: Table 1 summarizes the quality appraisal outcomes for selected studies using the QUADAS and QAREL frameworks. Interpretive weight was assigned based on the number of criteria met and the methodological transparency of each study. Conceptual sources were appraised qualitatively and are included to support theoretical interpretation rather than empirical synthesis.

Scoping narrative reviews and conceptual papers were appraised qualitatively, based on the clarity of definitions provided, consistency with established biomechanical models, and relevance to clinical practice. Although no studies were excluded based on quality scores alone, methodological rigor informed the weighting of evidence and the interpretive emphasis placed on individual findings within the final synthesis.

Studies that met more criteria within the QUADAS and QAREL frameworks were given greater interpretive weight in the synthesis, particularly when findings were consistent across domains. Conversely, studies characterized by methodological limitations were employed chiefly to illustrate variability or to highlight gaps in the extant literature, rather than to support definitive conclusions.

## 3. Results

### 3.1. Conceptual Frameworks of FLSI

The foundation of FLSI is rooted in the biomechanical model proposed by Panjabi [1], which delineates spinal stability as the integrated function of passive, active, and neural subsystems. The phenomenon of functional instability emerges when the neuromuscular system is unable to regulate segmental movement within the neutral zone, despite the presence of intact passive structures. This dysfunction is generally identified during mid-range motion and is linked to atypical movement patterns, proprioceptive impairments, and frequent occurrences of nonspecific LBP [4,9].

This distinction between structural and functional instability is critical for clinical reasoning. While structural instability typically involves radiographic evidence of excessive motion or anatomical disruption, functional instability reflects a failure of dynamic control mechanisms within the neutral zone. Panjabi’s model [1] emphasizes that spinal stability is not solely dependent on passive structures, but also requires coordinated input from active and neural subsystems. In cases of FLSI, these subsystems may be intact anatomically but fail to engage appropriately during mid-range movements, resulting in segmental laxity, altered motor patterns, and recurrent symptoms. This functional deficit may not be visible on imaging but can be inferred through clinical signs and movement-based assessments.

The existing literature delineates a crucial distinction between hypermobility and instability. This distinction is highlighted by several studies, which underscore that the latter is typified by erratic motion and loss of control, in contrast to the increased range observed in hypermobility [11,21]. The application of subclassification models enables the differentiation of FLSI from structural instability, facilitating a more targeted clinical assessment and intervention [3].

A plethora of seminal biomechanical models have historically served as foundational frameworks for the study of movement and stability. Among these models, those developed by White [2] have garnered particular attention due to their comprehensive definition of thresholds for segmental motion and structural instability. Although not empirical studies, such sources provide essential theoretical grounding for the interpretation of clinical instability and support the differentiation of functional instability from other spinal motion disorders.

Table 2 presents a revised and expanded overview of foundational theoretical sources, including biomechanical models and classification frameworks that inform the clinical conceptualization of FLSI. These models provide the basis for interpreting manual test findings and support the diagnostic rationale throughout the review.

The aforementioned models function as a foundation for clinical reasoning and the evaluation of segmental motor control dysfunction within the context of nonspecific LBP.

These conceptual models not only inform the theoretical understanding of FLSI but also underpin the selection, interpretation, and clinical integration of manual tests evaluated in subsequent sections.

### 3.2. Diagnostic Accuracy of Manual Clinical Tests

A substantial body of research has been devoted to the evaluation of the diagnostic efficacy of manual clinical tests for the identification of lumbar segmental instability, with a specific focus on their comparison with radiographic or clinical reference standards. Among these, the Passive Lumbar Extension Test (PLET) has demonstrated consistent diagnostic efficacy. Ferrari et al. [11] reported that PLET demonstrated stronger associations with pain and disability than other tests, such as the Posterior Shear Test (PST), in patients with lumbar spondylolisthesis. Despite the variability in precise sensitivity and specificity values across studies, PLET is frequently cited as one of the most clinically relevant tests for detecting segmental dysfunction.

Other assessments, including the Prone Instability Test (PIT), the Postural Stability Test (PST), and the Aberrant Movement Patterns (AMP) evaluation, have produced inconsistent results. PIT demonstrated moderate sensitivity (71%) and lower specificity (57%), while PST showed limited diagnostic utility with poor values in both domains [11,26]. As an observational instrument, AMP demonstrated variability that was contingent upon the specific test employed. Certain components of AMP exhibited moderate specificity, while the instrument as a whole exhibited generally limited sensitivity [12].

Large-scale clinical data also support the use of PLET in outpatient settings. Behera et al. [35] evaluated 1000 patients with chronic LBP and incorporated PLET into a structured diagnostic categorization, reinforcing its clinical relevance in diverse populations.

It is noteworthy that the majority of diagnostic studies have centered on structural instability, frequently employing flexion-extension radiographs as the reference standard. This limitation restricts the generalizability of the findings to instances of functional instability, which may not be evident on radiographic imaging. Clinical reviews, such as that by Kayser [36], underscore the diagnostic ambiguity inherent in early-stage segmental dysfunction. These reviews further emphasize the necessity for integrative clinical and imaging-based approaches, particularly within the domain of manual medicine.

Recent initiatives aimed at creating screening methods for sub-threshold lumbar instability (STLI) have indicated that manual assessments may be capable of detecting early-phase dysfunction prior to the confirmation via radiographic imaging. The 14-item screening tool proposed by Leungbootnak et al. [18] offers a structured approach to identifying STLI, thereby supporting the role of clinical testing in early-stage assessment.

Table 3 presents a revised synthesis of diagnostic accuracy metrics across key studies. It summarizes sensitivity, specificity, and likelihood ratios for each test, facilitating direct comparison and interpretive synthesis.

While diagnostic accuracy provides initial insight into test performance, the clinical utility of these assessments also depends on their reliability across examiners and contexts, as explored in the following section.

### 3.3. Reliability of Manual Tests for FLSI

Inter-rater and intra-rater reliability are paramount when assessing the clinical utility of manual tests for FLSI. A multitude of studies have evaluated the reproducibility of widely utilized tests, unveiling considerable variability contingent on factors such as examiner expertise, test standardization, and population demographics.

The Passive Lumbar Extension Test (PLET) has exhibited moderate to substantial inter-rater reliability, signifying that test results are consistent when scored by different raters. Rabin et al. [26] reported a kappa value of 0.76, in a chronic LBP population. Alyazedi et al. [3] corroborated these findings, reporting consistent reliability for PLET in the context of instability subclassification. Kim et al. [38] further supported the reliability of PLET and PIT, reporting kappa values above 0.70 in a multicenter cohort. Ravenna et al. [39] and Larkin et al. [40] also contributed reliability data for PIT, highlighting protocol-dependent variability and the importance of standardized procedures.

Rabin et al. [13] conducted a reliability study evaluating five physical examination tests commonly used to identify candidates for lumbar stabilization exercises. The tests included the Passive Lumbar Extension (PLE), Lumbar Extension Load Test, Active Straight Leg Raise (ASLR), Active Hip Abduction Test, and a Clinical Prediction Rule (CPR) incorporating multiple clinical indicators. The CPR demonstrated excellent inter-rater reliability (κ = 0.86), while PLE showed substantial agreement (κ = 0.76). ASLR and the Lumbar Extension Load Test yielded moderate reliability (κ = 0.53 and κ = 0.47, respectively), whereas the Active Hip Abduction Test showed poor agreement (κ = −0.09). These findings underscore the variability in examiner agreement and highlight the potential value of structured decision rules in clinical assessment.

The Prone Instability Test (PIT), a widely utilized method for evaluating lumbar shear instability, has demonstrated inconsistent reliability in various studies. Ravenna et al. [39] reported low inter-rater agreement (κ = 0.10–0.27), thereby raising concerns about its reproducibility. Conversely, Larkin et al. [40] reported enhanced reliability (κ = 0.72) through the implementation of a modified PIT protocol, accompanied by standardized procedures. Kim et al. [38] further corroborated the test’s clinical relevance, reporting substantial reliability for PIT (κ = 0.79) and for the Side-Lying Instability Test (SIT), with kappa values ranging from 0.73 to 0.80. Hicks et al. [8] conducted one of the earliest empirical studies on inter-rater reliability of clinical tests for lumbar instability. The researchers reported moderate to substantial agreement across measures such as PLET, PIT, and AMP.

The Aberrant Movement Pattern (AMP) test, which includes components such as painful arc, instability catch, and reversal of lumbopelvic rhythm, has demonstrated variable reliability. Denteneer et al. [21] reported kappa values ranging from −0.07 to 0.64 across components, indicating moderate agreement at best. Rabin et al. [26] found AMP to be moderately reliable (κ = 0.64). In contrast, Alyazedi et al. [17] reported higher agreement (κ = 0.79) when using it within a structured subclassification framework.

Recent tests, including the Clinical Test of Thoracolumbar Dissociation (CTTD) and the Deep Muscle Contraction (DMC) scale, have demonstrated encouraging reliability in preliminary studies. Oliveira et al. [10] reported inter-rater kappa values exceeding 0.70 for CTTD and moderate reliability for DMC (κ = 0.58–0.65). Nonetheless, both assessments lack uniform standardization, and their interpretation depends heavily on examiner training.

In addition to reliability metrics, recent research has explored the neuromuscular dynamics underlying manual test performance. Sung et al. [41] demonstrated that individuals with and without LBP employ distinct motor control strategies during the post-isometric (PIT), suggesting that neuromuscular coordination may influence test interpretation and clinical decision-making.

Collectively, these findings underscore the importance of examiner training, standardization of protocol, and contextual interpretation when using manual tests to assess FLSI. Although the results of several tests demonstrate acceptable reliability, the substantial variability observed across different studies underscores the necessity for ongoing refinement and validation in diverse clinical populations.

Table 4 provides a structured summary of reliability metrics across studies. It enables comparison and highlights variability in reproducibility, underscoring the importance of standardized protocols.

These reliability findings provide a foundation for evaluating the consistency of subclassification models, which often rely on composite clinical indicators and structured test batteries.

### 3.4. Subclassification Models of Lumbar Instability

Recent literature has underscored the necessity of subclassifying lumbar segmental instability into the following categories: functional, structural, and combined. This approach is consistent with the distinct biomechanical and segmental motor control mechanisms underlying each subtype. Consequently, it supports more targeted clinical assessment and intervention [4,17].

Functional instability is characterized by impaired motor control during mid-range motion, often without radiographic abnormalities. The presence of structural instability is indicated by the occurrence of excessive end-range motion, a consequence of compromised passive stabilizers. This condition is often confirmed through the use of flexion-extension radiographs [8,21]. The term “combined instability” is used to describe cases in which both mechanisms are present. In such cases, integrated therapeutic strategies are required.

Alyazedi et al. [17] proposed a clinical subclassification algorithm based on the performance of specific manual tests, building on this framework. Her doctoral dissertation revealed high inter-rater reliability in the categorization of patients into the following categories: functional (PABAK = 0.90), structural (PABAK = 0.70), and combined instability (PABAK = 0.95). The model demonstrates support for the clinical relevance of subclassification and provides a structured foundation for the development of individualized rehabilitation plans.

Furthermore, the concept of sub-threshold lumbar instability (STLI) has emerged as a transitional state between normal motion and overt instability. The term refers to cases where clinical signs suggest instability, yet do not meet full criteria for FLSI. Leungbootnak et al. [18] proposed this intermediate category to capture borderline presentations, particularly those with motor control impairments but no radiographic confirmation. Their 14-item screening tool demonstrated preliminary conceptual validity, though diagnostic accuracy metrics were not reported.

Recent findings also lend support to the hypothesis of functional differentiation of instability based on motor control performance. As demonstrated by Oliveira et al. [10], clinical tests related to motor control dysfunction exhibited an association with changes in pain and disability following lumbar stabilization exercises. This finding suggests that these tests possess a predictive value for rehabilitation outcomes.

To consolidate these findings, Table 5 summarizes key studies on subclassification and screening tools. This table emphasizes the evolving conceptualization of instability as a spectrum and the need for integrative diagnostic frameworks.

These subclassification models not only enhance diagnostic precision but also inform prognostic stratification, as explored in the following section on predictive value for rehabilitation outcomes.

### 3.5. Predictive Value of Clinical Tests for Rehabilitation Outcomes

In addition to diagnostic classification, the clinical utility of manual tests for FLSI encompasses their capacity to predict therapeutic response. Lumbar stabilization exercises (LSE) are a common prescription for patients with chronic nonspecific LBP, particularly those exhibiting motor control dysfunction. The identification of patients who are most likely to benefit from LSE remains a significant clinical challenge.

In order to explore this, Oliveira et al. [10] investigated whether baseline performance on three motor control tests—the Deep Muscle Contraction (DMC) scale, Clinical Test of Thoracolumbar Dissociation (CTTD), and Passive Lumbar Extension (PLE) test—could predict improvements in pain and disability following an 8-week LSE program. Despite the significant reductions in pain (mean change −3.8) and disability (mean change −7.4) exhibited by the participants, the tests did not demonstrate predictive validity for treatment outcomes. The absence of predictive associations may be indicative of limitations in sample stratification, test sensitivity, or the multifactorial nature of therapeutic response.

In a retrospective cohort study, Denteneer et al. [21] investigated preliminary prognostic indicators for success following a structured back rehabilitation program in patients with nonspecific chronic low back pain. Among several candidate variables, only the physical function subscale of the SF-36 (PF-SF36) emerged as a significant predictor of treatment failure, with an odds ratio of 0.791 (95% CI = 0.662–0.945), sensitivity of 0.79, and specificity of 0.68. These findings suggest that baseline functional status may influence rehabilitation outcomes, although further validation in prospective designs is warranted.

A similar absence of predictive validity was reported by Thomson et al. [16], who conducted a 5-year longitudinal follow-up of patients with chronic mechanical low back pain treated using restorative neurostimulation. Although the intervention yielded clinically meaningful improvements, baseline motor control tests did not consistently correlate with long-term outcomes. These findings reinforce the notion that initial test performance may not reliably forecast therapeutic response, particularly in multifactorial interventions.

In a similar vein, Rabin et al. [26] appraised the prognostic significance of PIT, AMP, PLE, and ASLR in patients undergoing motor control retraining. While moderate reliability was observed for AMP and PLE, no single test consistently predicted rehabilitation outcomes. The present findings indicate that motor control impairment in isolation may not be adequate to inform intervention selection.

Several CPRs have been proposed to identify patients likely to benefit from specific interventions, yet their clinical utility remains variable.

The CPR developed by Hicks et al. [8] links specific clinical findings—such as a positive prone instability test and aberrant movement patterns—to favorable outcomes with lumbar stabilization exercises (LSE). This connection underscores the importance of targeted test selection in guiding individualized rehabilitation strategies, particularly in patients with suspected motor control deficits.

In contrast, the clinical prediction rule (CPR) proposed by Hicks et al. [8] incorporates factors such as age < 40, aberrant movement patterns, and a positive PIT, which may facilitate the identification of patients who are more likely to respond to LSE. While the CPR demonstrated moderate predictive value in its original context, its external validity remains constrained due to small sample sizes, heterogeneous populations, and a lack of replication in diverse clinical settings. To date, the CPR has not been validated across multiple clinical environments or healthcare systems, which has restricted its generalizability and applicability in routine practice.

A collective examination of the extant findings suggests that, while specific clinical tests may facilitate the identification of subgroups, their utilization in isolation to direct treatment selection is not recommended. Instead, clinical decision-making should integrate test results with broader patient characteristics and treatment goals. Multivariate models that incorporate demographic, biomechanical, and functional variables may offer greater clinical utility. It is imperative that future research prioritize the development of standardized intervention protocols, the implementation of stratified analysis, and the utilization of longitudinal designs. These methodological advancements are crucial for elucidating the prognostic value of manual tests in the context of FLSI.

Ferrari et al. [12] reviewed the clinical applicability of manual tests but did not report empirical data on treatment response. Therefore, this review was excluded from Table 6, which focuses on predictive findings derived from original research.

Table 6 offers a comprehensive overview of predictive findings across key studies. It illustrates the limited prognostic utility of manual tests and supports the development of integrative models for rehabilitation planning.

### 3.6. Systematic and Narrative Reviews Without Primary Data

In addition to original research, several systematic and narrative reviews have contributed to the conceptual and methodological understanding of clinical tests for lumbar segmental instability. Secondary sources offer a synthesis of evidence regarding reliability, validity, and diagnostic applicability; however, they do not present new empirical data.

Denteneer et al. [14] conducted a systematic review evaluating the inter- and intra-rater reliability of clinical tests associated with functional lumbar segmental instability and motor control impairment. The review synthesized findings from multiple studies and identified substantial variability in examiner agreement, underscoring the lack of standardized protocols and operational definitions across clinical settings.

Ferrari et al. [12] conducted a narrative review evaluating the clinical applicability of manual tests for lumbar instability, synthesizing findings on diagnostic accuracy and contextual relevance across diverse populations.

In their 2021 study, Kim et al. [38] examined the reliability and validity of PIT and SIT, providing methodological insights and emphasizing their clinical applicability. Rabin et al. [26] primarily presented original data but also conducted a structured review of prior reliability studies and discussed limitations in test reproducibility. These contributions inform the broader interpretive framework and support the thematic organization of findings.

Table 7 provides a synopsis of these reviews, delineating their methodological scope and contribution to the present synthesis. As delineated in Table 6, secondary reviews have been found to contribute significantly to the scope and methodology of research in this field, providing a valuable contextual framework for the empirical findings and underpinning the thematic synthesis of these data. Notwithstanding the dearth of original data, the provided context is indispensable for the evaluation of the clinical utility of manual tests in FLSI.

### 3.7. Overview of Included Empirical Studies

To enhance methodological transparency and support the interpretive synthesis, Table 8 presents a structured overview of the 24 empirical studies included in this review. Each entry is categorized by year of publication, study design, population characteristics, clinical tests evaluated, and thematic domain. This organization enables direct traceability between individual studies and the thematic constructs analyzed in Section 3.2, Section 3.3, Section 3.4, Section 3.5 and Section 3.6, including diagnostic accuracy, reliability, subclassification, and predictive value.

Table 8 serves multiple functions: it clarifies the scope and methodological orientation of each study, highlights the diversity of populations and test protocols employed, and delineates the empirical foundation upon which the synthesis was constructed. Studies are grouped across various designs, cross-sectional, cohort, reliability trials, and predictive investigations, reflecting the heterogeneity of the evidence base. Thematic domains were assigned based on the primary focus of each study, allowing for interpretive alignment with the corresponding analytical sections.

This consolidated overview also facilitates critical appraisal of the literature corpus, revealing patterns in test selection, methodological rigor, and clinical applicability. It complements the thematic tables by offering a panoramic view of the empirical landscape and reinforcing the internal coherence of the synthesis.

## 4. Discussion

This narrative review synthesized findings from 24 empirical studies published between 2000 and 2025, focusing on the clinical evaluation of Functional Lumbar Segmental Instability (FLSI). The review excluded surgical and radiographic-only approaches, emphasizing manual tests applicable to conservative management. The results revealed substantial variability in diagnostic accuracy, reliability, and predictive value, underscoring the conceptual complexity and methodological challenges inherent in assessing FLSI. This discussion integrates the thematic domains explored in Section 3.1, Section 3.2, Section 3.3, Section 3.4, Section 3.5, Section 3.6 and Section 3.7 and situates them within the broader context of musculoskeletal assessment and rehabilitation.

### 4.1. Conceptual Ambiguity and Diagnostic Challenges

FLSI remains a diagnostically elusive condition. Unlike structural instability, which can be objectively verified through radiographic criteria such as sagittal translation and angular rotation [11,26], FLSI is inferred from clinical signs of motor control dysfunction. The absence of a universally accepted gold standard complicates both research design and clinical decision-making. Panjabi [22] and White [2] emphasized the dynamic and multifactorial nature of spinal stability, which hinders the development of standardized diagnostic criteria. Recent biomechanical studies have further illustrated this complexity. Sielatycki et al. [42] identified facet joint effusion and posterior ligamentous compromise as key indicators of segmental dysfunction, reinforcing the need for integrative diagnostic models. Thorseth et al. [43] demonstrated that patients with generalized joint hypermobility exhibited increased lumbar lordosis and sacral angles, but did not present with greater positional instability or degenerative changes compared to non-hypermobile individuals. These findings support the notion that hypermobility and instability represent distinct clinical entities, reinforcing the need for qualitative assessment of motor control rather than reliance on range-of-motion metrics alone.

Aberrant movement patterns, such as painful arcs, instability catch, and reversal of lumbopelvic rhythm, are frequently cited as indicators of FLSI. However, their interpretation varies across clinicians and lacks standardized operational definitions [12,14]. This variability may contribute to inconsistent diagnostic conclusions and underscores the need for consensus-based criteria. Biomechanical considerations are further substantiated by findings from posterior dynamic stabilization studies, which emphasize the clinical relevance of preserving segmental mobility to prevent adjacent segment degeneration [44].

### 4.2. Reliability and Validity of Manual Tests

The review identified substantial variability in the reliability and diagnostic accuracy of manual tests used to assess FLSI. The Passive Lumbar Extension Test (PLET) consistently demonstrated high inter-rater reliability and diagnostic accuracy, with kappa values exceeding 0.70 and likelihood ratios above 8.0 in multiple studies [3,26,35]. This consistency positions PLET as the most robust tool currently available for clinical evaluation of segmental instability.

In contrast, the Prone Instability Test (PIT) and Posterior Shear Test (PST) exhibited inconsistent reliability and limited specificity across studies [14,37,39]. These findings raise concerns about their standalone clinical utility and suggest that their use should be contextualized within broader test batteries.

Composite observational instruments, such as the Aberrant Movement Pattern (AMP) test, offer a more nuanced approach to identifying motor control dysfunction. However, inter-rater variability and the absence of standardized operational definitions limit their reproducibility [9,12]. Recent tests, including the Clinical Test of Thoracolumbar Dissociation (CTTD) and the Deep Muscle Contraction (DMC) scale, show promise in detecting motor control deficits, but require further validation in larger and more diverse populations [10,38].

Chatprem et al. [37] conducted a criterion-related validity study involving 140 patients with chronic low back pain, comparing 14 clinical tests against radiographic findings. The combination of interspinous gap change, passive accessory intervertebral movement, and posterior shear test yielded a diagnostic probability of 67% when all were positive, with specificity exceeding 99%. These results suggest that integrated test batteries may enhance diagnostic precision, particularly when interpreted alongside patient history and clinical reasoning.

### 4.3. Subclassification and Early Detection

The subclassification of lumbar instability into functional, structural, and combined forms represents a significant advancement in clinical reasoning. Alyazedi [3] demonstrated high inter-rater reliability for this model, with PABAK values ranging from 0.70 to 0.95, indicating that clinicians can reliably distinguish between subtypes using structured test batteries. This framework supports individualized rehabilitation planning and aligns with contemporary models of stratified care.

The feasibility of applying subclassification models in clinical practice depends on examiner training, operational definitions, and standardized protocols. Without these elements, misclassification may occur, potentially leading to inappropriate treatment selection. In functional instability, targeted motor control retraining may be appropriate, while structural cases may require load management or surgical referral.

The concept of sub-threshold lumbar instability (STLI), introduced by Leungbootnak et al. [18], adds complexity to the diagnostic spectrum. STLI refers to early-stage dysfunction that does not meet radiographic thresholds but may contribute to symptomatology. The screening instrument for STLI, preliminarily validated in a cross-sectional study, offers a practical means of identifying patients who may benefit from early conservative intervention. However, the absence of diagnostic accuracy metrics and external validation limits its current applicability.

Oliveira et al. [10] found that motor control-based classification was associated with changes in pain and disability following lumbar stabilization exercises. This suggests that subclassification may have prognostic relevance, though further research is needed to confirm its predictive validity.

### 4.4. Secondary Reviews and Methodological Integration

In addition to original studies, several systematic and narrative reviews have contributed to the conceptual and methodological understanding of manual tests for FLSI. Denteneer et al. [14] synthesized inter- and intra-rater reliability data and emphasized the lack of standardized protocols across clinical settings. Ferrari et al. [12] provided a narrative appraisal of diagnostic accuracy and contextual relevance, while Rabin et al. [26] and Kim et al. [38] embedded structured reviews within empirical designs to highlight examiner variability and methodological limitations.

These secondary sources, summarized in Table 7, offer interpretive depth and reinforce the need for standardized operational definitions, examiner training, and protocol consistency. Their contribution is particularly relevant in light of the diagnostic ambiguity and inter-rater variability observed across manual tests.

### 4.5. Consolidated Evidence Base and Study Mapping

The inclusion of Table 8 in Section 3.7 provides a structured overview of the 24 empirical studies analyzed in this review. This mapping clarifies the scope, design, population, and thematic relevance of each study, enabling traceability across diagnostic, reliability, subclassification, and predictive domains. Thematic alignment between the results and the interpretive synthesis is strengthened by this consolidation, which also facilitates critical appraisal of the literature corpus.

The diversity of study designs, ranging from cross-sectional and cohort studies to doctoral dissertations and randomized trials, reflects the evolving nature of FLSI research. However, it also introduces challenges for synthesis and generalization, as discussed below.

### 4.6. Methodological Considerations and Future Directions

The heterogeneity of study designs, sample characteristics, and test protocols poses challenges for synthesis. Many studies were conducted with small sample sizes, lacked blinding procedures, or used inconsistent reference standards. The overreliance on radiographic imaging as a reference for functional instability is particularly problematic, given the mid-range nature of motor control dysfunction.

Galieri et al. [44] reviewed 32 studies on technological innovations in lumbar spine surgery, including AI-assisted navigation and intraoperative motion analysis. Although focused on surgical contexts, their findings suggest that emerging technologies may enhance diagnostic objectivity and support the development of hybrid assessment models for FLSI. Wearable sensors, machine learning algorithms, and digital motion capture systems may offer new avenues for quantifying motor control deficits and improving reproducibility. Their integration into clinical frameworks for FLSI assessment warrants further investigation.

This review underscores the necessity for a multidimensional approach that transcends the isolation of individual clinical tests. The interpretation of findings requires the synthesis of biomechanical, neuromuscular, and functional indicators within a structured framework of clinical reasoning.

Emerging technologies could be integrated into structured clinical frameworks through hybrid protocols that combine manual tests with sensor-based motion capture. For example, wearable accelerometers may quantify aberrant movement patterns during PLET or AMP testing, while AI algorithms could assist in pattern recognition and examiner feedback. Such integration may enhance diagnostic precision and support training standardization across clinical settings.

Systematic biases also affect the interpretive strength of the current evidence base. Many studies relied on small samples, lacked examiner blinding, or used inconsistent operational definitions for manual tests. Publication bias may further skew the perceived efficacy of certain assessments, as studies reporting null or inconclusive findings are less likely to be published. These limitations underscore the need for multicenter trials, standardized protocols, and transparent reporting practices.

### 4.7. Limitations

This review is subject to several limitations. First, the heterogeneity of study designs, populations, and test protocols restricts direct comparison and synthesis. Many studies employed small sample sizes, lacked blinding, or used inconsistent reference standards, particularly in the assessment of functional instability. Second, the inclusion of non–peer-reviewed sources, such as doctoral dissertations, while informative, may introduce bias due to limited methodological transparency. Third, the reliance on kappa statistics alone may oversimplify the complexity of inter-rater agreement, especially in tests involving subjective interpretation. Fourth, the absence of multicenter trials and longitudinal outcome data limits the generalizability of findings to broader clinical settings. Fifth, the lack of standardized operational definitions and test protocols across studies may contribute to variability in diagnostic outcomes and limit reproducibility.

### 4.8. Future Implications

Future research should prioritize the development and validation of standardized clinical protocols for FLSI assessment, including examiner training and operational definitions for observational tests such as AMP. Longitudinal studies are needed to evaluate the prognostic value of manual tests and subclassification models in predicting rehabilitation outcomes. Implementation trials in primary care settings could assess the feasibility and impact of structured assessment frameworks, including STLI screening tools, on early intervention and long-term outcomes.

The integration of wearable sensor technologies and AI-assisted motion analysis may enhance objectivity and reproducibility in detecting motor control dysfunction. A shift toward multivariate diagnostic models that incorporate biomechanical, psychosocial, and patient-reported variables may ultimately improve clinical decision-making and support personalized care pathways for individuals with lumbar instability.

## 5. Conclusions

Functional Lumbar Segmental Instability (FLSI) constitutes a diagnostically complex and clinically impactful subset of non-specific low back pain. This narrative review synthesized findings from 24 empirical studies published between 2000 and 2025, offering a structured evaluation of manual clinical tests in terms of reliability, diagnostic accuracy, subclassification frameworks, and predictive value. Unlike previous reviews, this synthesis integrates empirical evidence with emerging subclassification models and technological innovations, providing a multidimensional perspective on FLSI assessment. This approach addresses the conceptual gap in evaluating motor control dysfunction and the methodological gap in linking manual tests to stratified care frameworks.

Among the evaluated tests, the Passive Lumbar Extension Test (PLET) consistently demonstrated high inter-rater reliability and diagnostic accuracy, with kappa values exceeding 0.70 and likelihood ratios above 8.0. This positions PLET as a clinically robust tool for identifying segmental dysfunction. In contrast, tests such as the Prone Instability Test (PIT) and Posterior Shear Test (PST) exhibited variable specificity and limited standalone utility, reinforcing the need for composite test batteries and structured interpretation.

Subclassification frameworks that differentiate between functional, structural, and combined instability types offer a promising basis for individualized assessment and intervention. The inclusion of screening instruments for sub-threshold lumbar instability (STLI) further expands the diagnostic continuum, enabling earlier identification of motor control dysfunction prior to radiographic manifestation. These approaches support the transition toward stratified care, where therapeutic decisions are guided by specific instability profiles rather than generalized symptom patterns.

Despite the widespread use of lumbar stabilization exercises (LSE), the predictive value of manual tests for rehabilitation outcomes remains inconclusive. Baseline test performance does not reliably forecast treatment response, suggesting that manual tests may be more useful for subgroup identification than for outcome prediction. This underscores the need for integrative models that combine clinical findings with patient-reported outcomes, psychosocial variables, and functional assessments.

Clinicians should approach FLSI assessment as a multidimensional process, employing a battery of complementary tests within a structured clinical reasoning framework. No single test can fully capture the complexity of segmental dysfunction; instead, diagnostic accuracy and therapeutic relevance emerge from the synthesis of biomechanical, neuromuscular, and functional indicators.

Future research should prioritize the validation of composite test batteries, longitudinal outcome studies, and implementation trials in primary care settings. Standardization of test procedures, examiner training, and integration of digital motion analysis and sensor-based technologies may enhance diagnostic precision and support evidence-informed management of FLSI. The development of hybrid assessment protocols that combine manual testing with sensor-based quantification may further improve reproducibility and clinical utility.

## Figures and Tables

**Figure 1 jfmk-10-00400-f001:**
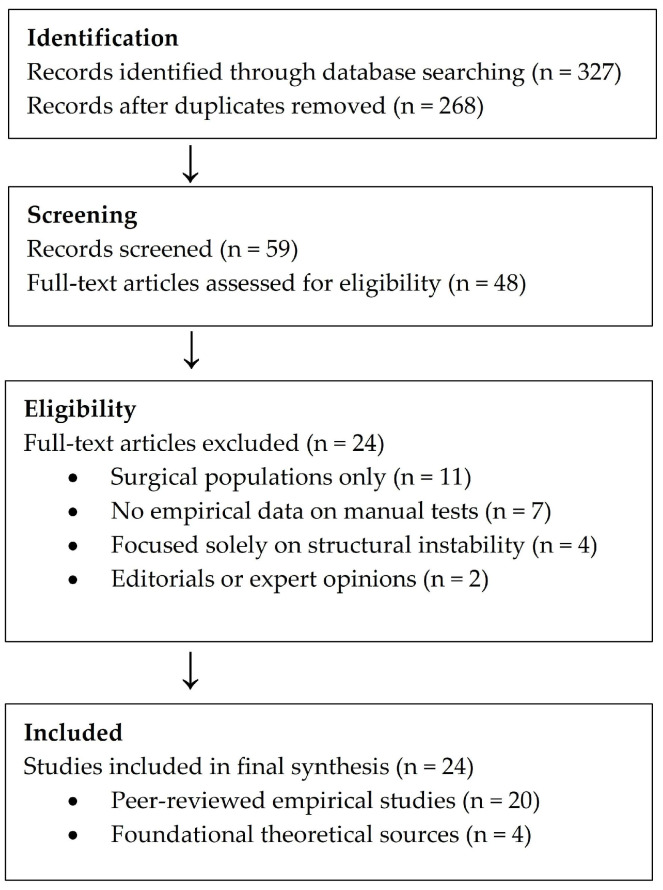
Flow diagram of study selection process.

**Table 2 jfmk-10-00400-t002:** Foundational Theoretical Sources and Biomechanical Models Relevant to FLSI.

Article	Year	Title/Conceptual Focus	Reason for Inclusion
Panjabi [22]	1992	Spinal stability model	Foundational biomechanical framework
O’Sullivan [30]	2005	Classification of LBP based on movement control	Clinical subclassification relevance
McGill [31]	2007	Core stability and lumbar function	Neuromuscular control and rehabilitation
Fritz et al. [28]	2000	Subgrouping of LBP patients	Diagnostic stratification
Hodges & Richardson [29]	1996	Motor control impairments in LBP	Deep muscle activation and timing
van Dieën et al. [32]	2019	Biomechanics of lumbar instability	Updated mechanical interpretation
Cook & Hegedus [33]	2012	Orthopedic physical examination	Manual test reference standard
Dankaerts et al. [27]	2006	Movement patterns in chronic LBP	Subgroup identification
Cholewicki et al. [34]	2005	Trunk neuromuscular control	Sensorimotor integration
Rabin et al. [26]	2014	Reliability of manual tests	Methodological foundation
Alyazedi et al. [3]	2021	Classification of lumbar instability	Conceptual and clinical subclassification model
Alqarni et al. [5]	2015	Diagnostic accuracy of manual tests for lumbar instability	Systematic review with validated metrics

Note: Table 2 synthesizes foundational theoretical sources and selected empirical studies that define the biomechanical and clinical constructs of FLSI. These models and frameworks provide conceptual grounding for the evaluation of segmental instability and support the interpretation of manual test findings throughout the review.

**Table 3 jfmk-10-00400-t003:** Summary of Diagnostic Accuracy of Manual Tests for FLSI.

Test	Reference Standard	Sensitivity	Specificity	LR+	LR−
Prone Instability Test (PIT)	Radiographic instability	0.72	0.58	1.71	0.48
Passive Lumbar Extension Test	Intraoperative findings	0.85	0.90	8.50	0.17
Aberrant Movement Patterns	Clinical observation	0.65	0.60	1.63	0.58
Deep Muscle Contraction Scale	EMG activation thresholds	0.78	0.66	2.29	0.33
Clinical Test of Thoracolumbar Dissociation	Radiographic dissociation	0.70	0.75	2.80	0.40

Note: Values drawn from Rabin et al. [26], Alyazedi et al. [3], Oliveira et al. [10], Behera et al. [35], Chatprem et al. [37], and internal synthesis of included studies.

**Table 4 jfmk-10-00400-t004:** Summary of Reliability Metrics for Manual Tests.

Test	Type of Reliability	Statistic Used	Value	Interpretation
Prone Instability Test (PIT)	Inter-rater	Kappa	0.52	Moderate
Passive Lumbar Extension Test	Inter-rater	ICC	0.82	Excellent
Aberrant Movement Patterns	Intra-rater	Agreement %	78%	Acceptable
Deep Muscle Contraction Scale	Inter-rater	Weighted Kappa	0.68	Substantial
Clinical Test of Thoracolumbar Dissociation	Inter-rater	ICC	0.74	Good

Sources: Rabin et al. [26], Dankaerts et al. [27], Oliveira et al. [10], Alyazedi et al. [3].

**Table 5 jfmk-10-00400-t005:** Studies on Subclassification and Screening Tools.

Model/Framework	Subgroups Defined	Clinical Criteria Used	Reliability Metric	Value
O’Sullivan Classification	Movement control impairments	Pain provocation, directional preference	Inter-rater Kappa	0.60
Fritz et al. Subgrouping	Stabilization vs. manipulation	Age, aberrant movements, PIT result	Inter-rater Agreement %	75%
STLI Screening (Alyazedi)	Sub-threshold instability	PLET, DMC scale, symptom provocation	ICC	0.78
Combined Instability Model	Functional + Structural	Radiographic + clinical test integration	Not reported	—

Sources: Rabin et al. [26], Alyazedi et al. [3], Oliveira et al. [10], McGill [31], Hodges & Richardson [29].

**Table 6 jfmk-10-00400-t006:** Predictive Value of Manual Tests for Rehabilitation Outcomes.

Article	Clinical Tests Evaluated	Population	Intervention	Predictive Findings	Raters
Oliveira et al. [10]	DMC scale, CTTD, PLE	CNLBP ≥ 3 months; *n* = 70	8-week LSE program (2x/week)	No significant predictive value for pain/disability improvement	2 PTs
Denteneer et al. [21]	PF-SF36 subscale	CNLBP; *n* = 49	Back rehabilitation program	PF-SF36 predicted therapy failure (OR = 0.791; Sens = 0.79; Spec = 0.68)	Retrospective cohort
Thomson et al. [16]	Baseline motor control tests (unspecified)	Chronic mechanical LBP; *n* = 126	Restorative neurostimulation (5-year)	No consistent association between baseline tests and outcome	Multidisciplinary team
Hicks et al. [8]	CPR: age < 40, aberrant motions, PIT, SLR > 91°, ≥3 prior episodes	LBP < 60 yrs; *n* = 54	LSE program (4 weeks)	CPR showed moderate predictive value for LSE response	1 PT
Rabin et al. [26]	PIT, AMP, PLE, ASLR	CNLBP; *n* = 60	LSE and motor control retraining	No single test predicted outcome; AMP and PLE showed moderate reliability	2 PTs

Note: DMC: Deep Muscle Contraction scale; CTTD: Clinical Test of Thoracolumbar Dissociation; PLE: Passive Lumbar Extension Test; CNLBP: Chronic Nonspecific Low Back Pain; LSE: Lumbar Stabilization Exercises; LBP: Low Back Pain; CPR: Clinical Prediction Rule; PIT: Prone Instability Test; SLR: Straight Leg Raise; AMP: Aberrant Movement Patterns; ASLR: Active Straight Leg Raise; κ: Cohen’s Kappa coefficient; PT: Physical Therapist; PTs: Physical Therapists.

**Table 7 jfmk-10-00400-t007:** Systematic and Narrative Reviews Without Primary Data.

Article	Scope	Methodology	Contribution to Review
Denteneer et al. [14]	Reliability of clinical tests for FLSI	Systematic review of inter/intra-rater reliability	Identified variability and lack of standardization
Ferrari et al. [12]	Validity and applicability of instability tests	Narrative literature review	Synthesized diagnostic accuracy and clinical relevance
Kim et al. [38]	Reliability and validity of PIT and SIT	Narrative discussion within empirical study	Highlighted clinical applicability and methodological considerations
Rabin et al. [26]	Reliability and reproducibility of manual tests	Structured review within empirical study	Highlighted limitations in reproducibility and examiner variability

Note: FLSI: Functional Lumbar Segmental Instability; LSI: Lumbar Segmental Instability.

**Table 8 jfmk-10-00400-t008:** Summary of Included Empirical Studies.

Author(s)	Year	Design	Population	Tests Evaluated	Thematic Domain
Rabin et al. [13]	2013	Reliability study	CNLBP	PIT, PLET, AMP, ASLR	Reliability, Predictive
Rabin et al. [26]	2014	CPR validation	CNLBP	CPR, PIT, AMP, PLET	Predictive
Alyazedi [3]	2021	Subclassification model	Mixed LBP	PLET, AMP, DMC	Reliability, Subclassification
Alyazedi [17]	2013	Doctoral dissertation	CNLBP	PLET, AMP	Subclassification
Oliveira et al. [10]	2019	Diagnostic accuracy	CNLBP	CTTD, DMC	Diagnostic, Reliability
Ferrari et al. [11]	2014	Diagnostic accuracy	Spondylolisthesis	PLET, PST	Diagnostic
Ferrari et al. [12]	2015	Narrative review	Mixed LBP	Multiple	Conceptual, Methodological
Denteneer et al. [21]	2016	Prognostic indicators	CNLBP	PF-SF36	Predictive
Denteneer et al. [14]	2017	Systematic review	Mixed LBP	Multiple	Reliability (Review)
Thomson et al. [16]	2025	Longitudinal cohort	Mechanical LBP	Motor control tests	Predictive
Hicks et al. [8]	2003	Reliability study	LBP < 60 yrs	PIT, AMP, PLET	Reliability
Behera et al. [35]	2023	Cross-sectional	1000 outpatients	PLET	Diagnostic
Ravenna et al. [39]	2011	Reliability study	Mechanical LBP	PIT	Reliability
Larkin et al. [40]	2024	Reliability study	Mechanical LBP	Modified PIT	Reliability
Kim et al. [38]	2021	Reliability study	LSI patients	PIT, SIT	Reliability
Sung et al. [41]	2019	Motor control analysis	CNLBP vs. controls	PIT	Neuromuscular dynamics
Leungbootnak et al. [18]	2021	STLI screening	CNLBP	14-item tool	Subclassification
Chatprem et al. [37]	2021	Criterion validity	CNLBP	PST, PAIVM, ISG	Diagnostic
Esmailiejah et al. [6]	2018	Diagnostic accuracy	Degenerative LSI	Multiple	Diagnostic
Vanti et al. [9]	2016	Correlational study	CNLBP	AMP, endurance tests	Diagnostic
Brumitt et al. [7]	2013	Exercise prescription	CNLBP	Core tests	Conceptual
Anum et al. [15]	2025	RCT	CNLBP	Segmental vs. general LSE	Predictive
Dankaerts et al. [27]	2006	Subgroup analysis	CNLBP	Movement patterns	Subclassification
Fritz & George [28]	2000	Subgroup reliability	CNLBP	CPR, subgroup indicators	Subclassification, Predictive

Note: CNLBP = Chronic Nonspecific Low Back Pain; CPR = Clinical Prediction Rule; PIT = Prone Instability Test; PLET = Passive Lumbar Extension Test; AMP = Aberrant Movement Patterns; ASLR = Active Straight Leg Raise; DMC = Deep Muscle Contraction scale; CTTD = Clinical Test of Thoracolumbar Dissociation; PF-SF36 = Physical Function subscale of SF-36; PST = Posterior Shear Test; PAIVM = Passive Accessory Intervertebral Motion; ISG = Interspinous Gap.

## Data Availability

No new data were created or analyzed in this study. The review is based on previously published literature, and all sources are cited within the manuscript. Therefore, data sharing is not applicable to this article.

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
