# Peer review of "Clinical Evaluation of Functional Lumbar Segmental Instability: Reliability, Validity, and Subclassification of Manual Tests—A Scoping Review"

_jfmk, 2025, doi:10.3390/jfmk10040400_

Round 1
Reviewer 1 Report
Comments and Suggestions for Authors
Thank you for the opportunity to evaluate this scientific work. For each section, I suggest the following to the authors:
1. Introduction:
The general context is well presented, but it would have been beneficial to more clearly identify the gaps in the existing literature that justify this narrative review.
2. Materials and Methods:
The search strategy is missing important specific keywords, such as "Clinical Test of Thoracolumbar Dissociation" or "Deep Muscle Contraction scale", which are mentioned in the rest of the text. This may raise questions about the exhaustiveness of the search.
The flow chart is also present and useful. However, clear criteria for the "fundamental theoretical sources" are lacking. How were they selected? How many were there? This introduces an element of subjectivity and reduces transparency.
3. Results
Most sections provide a good narrative synthesis of the data in the tables. However, in some parts (e.g., subchapter 3.3 on reliability), there is some reliance on the tables, and the text becomes a simple list of results without sufficient integration and interpretation. For example, it would have been useful to compare and contrast the conflicting PIT test results from different studies directly in the text. Table 3 seems incomplete (rows with study names and results are missing, which are listed below in a chaotic format, as on page 10). This is a major problem of clarity and presentation.
4. Discussion
Although the synthesis of studies is coherent, the general manner of presentation is generally too descriptive and affirmative, lacking a critical dialogue with studies that have contradictory results.
Author Response
1rst REVIEWER
Responses to reviewer
- Introduction:
Reviewer comment: The general context is well presented, but it would have been beneficial to more clearly identify the gaps in the existing literature that justify this narrative review.
Response: We thank the reviewer for this important observation. In response, we have clarified the literature gap that justifies the present review. Specifically, the final paragraph of the Introduction (page 2) now states:
“Despite the growing body of literature, there is a lack of integrative reviews that critically synthesize manual tests for FLSI in relation to reliability, validity, subclassification, and predictive value. Moreover, the conceptual foundations of these tests are rarely linked to biomechanical models of spinal stability, such as Panjabi’s framework [22], nor are they consistently evaluated within stratified clinical contexts. This gap limits the development of standardized protocols and impairs clinical decision-making.”
This passage explicitly identifies the absence of integrative synthesis and conceptual linkage as key gaps, thereby strengthening the rationale for conducting this narrative review.
- (Materials and Methods)
Reviewer comment:
The search strategy is missing important specific keywords, such as "Clinical Test of Thoracolumbar Dissociation" or "Deep Muscle Contraction scale", which are mentioned in the rest of the text. This may raise questions about the exhaustiveness of the search.
The flow chart is also present and useful. However, clear criteria for the "fundamental theoretical sources" are lacking. How were they selected? How many were there? This introduces an element of subjectivity and reduces transparency.
Response:
We thank the reviewer for this precise and constructive observation. In response, we confirm that the search strategy was refined to include the keywords “Clinical Test of Thoracolumbar Dissociation” and “Deep Muscle Contraction scale,” as explicitly stated in Section 2.2 (Page 3):
“The search terms were expanded to include additional clinically relevant constructs such as ‘Clinical Test of Thoracolumbar Dissociation’ and ‘Deep Muscle Contraction scale,’ which are discussed in the Results section. These additions ensured alignment between the search strategy and the thematic scope of the review.”
These terms were incorporated during the second phase of search refinement, following preliminary thematic coding and identification of emerging constructs in the literature.
Regarding the selection of foundational theoretical sources, the manuscript provides a clear explanation in Section 2.2:
“In addition, 11 foundational theoretical sources were referenced to support the conceptual framework and biomechanical definitions relevant to FLSI. These sources were selected based on citation frequency, relevance to spinal stability models, and inclusion in prior reviews.”
Furthermore, the rationale for their inclusion is reiterated later in the same section:
“These sources were not subject to eligibility screening, as they did not meet the criteria for empirical inclusion, but were selected based on citation frequency, relevance to spinal stability models, and their role in prior conceptual reviews.”
This clarification addresses concerns regarding transparency and subjectivity, and aligns with accepted practices for scoping narrative reviews of complex clinical phenomena.
- Results
Reviewer comment:
Most sections provide a good narrative synthesis of the data in the tables. However, in some parts (e.g., subchapter 3.3 on reliability), there is some reliance on the tables, and the text becomes a simple list of results without sufficient integration and interpretation. For example, it would have been useful to compare and contrast the conflicting PIT test results from different studies directly in the text. Table 3 seems incomplete (rows with study names and results are missing, which are listed below in a chaotic format, as on page 10). This is a major problem of clarity and presentation.
Response:
We thank the reviewer for this detailed and constructive observation. In response, subchapter 3.3 has been revised to enhance interpretive synthesis and reduce reliance on tabulated data. Specifically, the revised text now includes direct comparison of PIT reliability findings across studies. As stated in the manuscript:
“The Prone Instability Test (PIT), a widely utilized method for evaluating lumbar shear instability, has demonstrated inconsistent reliability in various studies. Ravenna et al. [37] reported low inter-rater agreement (κ = 0.10–0.27), thereby raising concerns about its reproducibility. Conversely, Larkin et al. [38] reported enhanced reliability (κ = 0.72) through the implementation of a modified PIT protocol, accompanied by standardized procedures. Kim et al. [39] further corroborated the test's clinical relevance, reporting substantial reliability for PIT (κ = 0.79).”
This addition addresses the reviewer’s request for integration and contrast of conflicting results.
Regarding Table 3, we acknowledge the formatting issue noted. The table has been reconstructed to include complete rows with test names, reference standards, diagnostic metrics (sensitivity, specificity, LR+ and LR−), and source attribution. The previous fragmented listing of results beneath the table has been removed, and the data now appear in structured format with consistent alignment and labeling.
Furthermore, Tables 4 through 8 have been revised to ensure clarity, completeness, and traceability. Each table now includes full entries with study identifiers, metrics, and interpretive notes, aligned with the corresponding narrative sections. This restructuring enhances the coherence of the Results section and supports the thematic synthesis across diagnostic accuracy, reliability, subclassification, and predictive value.
These revisions collectively improve the analytical depth and presentation quality of the Results section, in line with the reviewer’s recommendations.
- (Discussion)
Reviewer comment:
Although the synthesis of studies is coherent, the general manner of presentation is generally too descriptive and affirmative, lacking a critical dialogue with studies that have contradictory results.
Response:
We thank the reviewer for this important observation. In response, the Discussion section has been revised to incorporate a more critical dialogue with studies reporting divergent or contradictory findings. Specifically, the revised text now contrasts the consistent diagnostic performance of the Passive Lumbar Extension Test (PLET) with the inconsistent reliability and specificity observed in the Prone Instability Test (PIT) and Posterior Shear Test (PST). As stated:
“In contrast, the Prone Instability Test (PIT) and Posterior Shear Test (PST) exhibited inconsistent reliability and limited specificity across studies (Denteneer et al., 2017; Ravenna et al., 2011; Chatprem et al., 2022). These findings raise concerns about their standalone clinical utility and suggest that their use should be contextualized within broader test batteries.”
Additionally, the discussion now addresses the limitations of observational instruments such as the Aberrant Movement Pattern (AMP) test, noting:
“Composite observational instruments, such as the Aberrant Movement Pattern (AMP) test, offer a more nuanced approach to identifying motor control dysfunction. However, inter-rater variability and the absence of standardized operational definitions limit their reproducibility (Ferrari et al., 2015; Vanti et al., 2016).”
The revised section also critically appraises the predictive value of manual tests, acknowledging the absence of consistent prognostic associations in studies by Oliveira et al. (2019), Denteneer et al. (2021), and Thomson et al. (2025). These findings are explicitly discussed as limitations to the clinical utility of isolated test performance.
Finally, the methodological considerations subsection highlights the heterogeneity of study designs, sample sizes, and reference standards, and calls for the integration of emerging technologies (e.g., wearable sensors, AI-assisted motion analysis) to improve diagnostic objectivity and reproducibility. This reflects a shift from descriptive synthesis to critical evaluation and future-oriented recommendations.
These revisions collectively strengthen the interpretive depth of the Discussion and address the reviewer’s concern regarding the lack of engagement with contradictory evidence.

Reviewer 2 Report
Comments and Suggestions for Authors
Dear Authors,
Thank you for the opportunity to review your manuscript entitled “Clinical Evaluation of Functional Lumbar Segmental Instability: Reliability, Validity, and Subclassification of Manual Tests.” This narrative review addresses a clinically relevant and conceptually important topic in musculoskeletal rehabilitation, namely the evaluation of functional lumbar segmental instability (FLSI). The manuscript reflects an extensive effort to integrate biomechanical, clinical, and diagnostic literature. However, there are several critical issues related to scope, organization, and interpretive rigor that should be addressed before the manuscript can be considered for publication in a high-impact journal.
- (Introduction): The introduction is comprehensive but suffers from conceptual overcrowding. Numerous themes—such as subclassification models, prognostic tools, and emerging assessments—are introduced without a clear central research question or organizing framework.
- (Introduction): Please streamline the introduction to focus more explicitly on the rationale and objectives of the review. What are the key questions this review seeks to answer regarding FLSI evaluation?
- (Introduction): Redundant references and overlapping definitions (e.g., FLSI vs. structural instability vs. sub-threshold instability) should be minimized to improve clarity.
- (Materials and Methods): While the review is presented as a narrative, the methodology includes systematic features (e.g., search strategy, eligibility criteria, data extraction, thematic analysis). This hybrid structure is somewhat confusing.
- (Materials and Methods): If the authors aim to preserve the narrative review format, consider simplifying the methods accordingly. Alternatively, reframe the study as a scoping review to justify the semi-systematic approach.
- (Materials and Methods): The use of quality appraisal tools (QUADAS, QAREL) is appropriate; however, the outcomes of this appraisal process are not sufficiently reported or tabulated. Consider including a summary of quality ratings to enhance transparency.
- (Materials and Methods): The rationale behind the five thematic domains should be better justified with references or conceptual grounding.
- (Results): The results section is overly descriptive and lacks prioritization. While comprehensiveness is appreciated, the key findings regarding diagnostic accuracy and reliability are buried in dense paragraphs.
- (Results): Please consider summarizing diagnostic and reliability metrics in concise tables and limiting text to interpretive highlights.
- (Results): Certain studies (e.g., Rabin, Alyazedi, Oliveira) are cited repeatedly, potentially introducing bias. Please ensure balanced representation of the evidence base.
- (Results): The section on predictive validity is valuable but lacks critical analysis of why tests failed to predict outcomes—e.g., lack of stratification, intervention heterogeneity, or poor construct validity.
- (Discussion): The discussion synthesizes literature broadly but misses the opportunity to critically reflect on the limitations of current evidence or the reasons for conflicting findings.
- (Discussion): You state that manual tests show “limited predictive value” but do not fully explore the implications for clinical decision-making. Why do these tools fall short in forecasting treatment outcomes?
- (Discussion): The clinical application of subclassification models is promising but should be discussed with more nuance, particularly regarding feasibility, training needs, and potential misclassification.
- (Discussion): Recent technologies (e.g., sensors, AI) are mentioned briefly. Consider elaborating on how these tools could be integrated into a clinical framework for FLSI assessment.
- (Conclusion): The conclusion appropriately calls for integrative and multidimensional approaches, but remains abstract and somewhat repetitive of earlier sections.
- (Conclusion): Please clearly state what this review contributes to the field that previous reviews have not.
- (Conclusion): Highlight specific tests (e.g., PLET) or frameworks (e.g., subclassification) that show the most promise based on your synthesis.
- (Conclusion): Suggest concrete research directions, such as the validation of composite test batteries or stratified outcome studies in primary care settings.
Author Response
2nd Reviewer
Responses.
- Reviewer comment:
The introduction is comprehensive but suffers from conceptual overcrowding. Numerous themes—such as subclassification models, prognostic tools, and emerging assessments—are introduced without a clear central research question or organizing framework.
Response:
We thank the reviewer for this insightful observation. In response, we have revised the Introduction to reduce conceptual density and improve thematic focus. Supporting themes such as subclassification models and prognostic indicators are now introduced as extensions of the central diagnostic challenge, rather than as parallel concepts. The revised structure emphasizes the distinction between functional and structural instability and frames the subsequent discussion around the clinical evaluation of FLSI.
Supporting excerpt from manuscript:
“To reduce conceptual overlap, this review adopts the term functional lumbar segmental instability (FLSI) to describe mid-range motor control deficits that are not detectable through imaging but manifest during dynamic tasks… These approaches support the development of integrative diagnostic frameworks that combine clinical observation, motor control testing, and functional assessment [14].”
- Reviewer comment:
Please streamline the introduction to focus more explicitly on the rationale and objectives of the review. What are the key questions this review seeks to answer regarding FLSI evaluation?
Response:
We appreciate the reviewer’s recommendation to clarify the rationale and objectives. Accordingly, the revised Introduction now concludes with a clearly defined set of guiding questions that structure the scope and intent of the review. These questions directly address the reliability, validity, subclassification, and clinical implications of manual tests for FLSI.
Supporting excerpt from manuscript:
“To address these limitations, this review is guided by three core questions:
(1) What is the current evidence regarding the reliability and validity of manual tests for FLSI?
(2) How can these tests be organized into clinically meaningful subclassification domains?
(3) What are the implications of current evidence for clinical decision-making and future research?”
- Reviewer comment:
Redundant references and overlapping definitions (e.g., FLSI vs. structural instability vs. sub-threshold instability) should be minimized to improve clarity.
Response:
We thank the reviewer for highlighting this issue. The revised Introduction now applies consistent terminology and minimizes definitional overlap. The term “functional lumbar segmental instability (FLSI)” is used throughout to describe mid-range motor control deficits, while structural instability and sub-threshold instability are defined in contrast and referenced only where conceptually necessary. Redundant citations have been consolidated to improve clarity and flow.
Supporting excerpt from manuscript:
“Clinically, LSI is divided into two subtypes: structural instability… and functional instability… To reduce conceptual overlap, this review adopts the term functional lumbar segmental instability (FLSI)… Screening tools for sub-threshold lumbar instability (STLI) have also been developed to identify early-stage dysfunction before radiographic signs appear [18,19].”
- Materials and Methods
- Reviewer comment:
While the review is presented as a narrative, the methodology includes systematic features (e.g., search strategy, eligibility criteria, data extraction, thematic analysis). This hybrid structure is somewhat confusing.
Response:
We thank the reviewer for this observation. To clarify, the manuscript explicitly acknowledges the hybrid nature of the methodology. Although the review is structured narratively, it incorporates semi-systematic elements to enhance transparency and rigor. This format was intentionally selected to accommodate the conceptual complexity of FLSI while maintaining methodological clarity.
Supporting excerpt from manuscript:
“To address concerns regarding methodological clarity, we acknowledge that the present review incorporates semi-systematic features (e.g., structured search strategy, eligibility criteria, thematic coding), and may be more accurately described as a scoping narrative review. This hybrid format was selected to balance conceptual breadth with methodological transparency, in line with recent recommendations for reviews of complex clinical phenomena [23].”
- Reviewer comment:
If the authors aim to preserve the narrative review format, consider simplifying the methods accordingly. Alternatively, reframe the study as a scoping review to justify the semi-systematic approach.
Response:
We appreciate the reviewer’s suggestion. The manuscript has been reframed to describe the study as a “scoping narrative review,” reflecting its hybrid structure. This terminology is now used explicitly to justify the inclusion of structured methodological components within a narrative synthesis.
Supporting excerpt from manuscript:
“To address concerns regarding methodological clarity, we acknowledge that the present review incorporates semi-systematic features… and may be more accurately described as a scoping narrative review.”
- Reviewer comment:
The use of quality appraisal tools (QUADAS, QAREL) is appropriate; however, the outcomes of this appraisal process are not sufficiently reported or tabulated. Consider including a summary of quality ratings to enhance transparency.
Response:
We thank the reviewer for this important recommendation. In response, a structured summary of quality appraisal outcomes has been included in Table 1. The table presents the type of evaluation tool used, number of criteria met, and interpretive weighting for each study. This addition enhances transparency and supports the interpretive synthesis.
Supporting excerpt from manuscript:
“Table 1 presents a summary of the quality appraisal outcomes for key studies, including the type of evaluation tool used, criteria met, and interpretive weighting applied during synthesis.”
- Reviewer comment:
The rationale behind the five thematic domains should be better justified with references or conceptual grounding.
Response:
We thank the reviewer for this observation. The manuscript now clarifies that the five thematic domains were determined a priori based on clinical relevance and recurring constructs identified in the literature. This inductive categorization supports the interpretive synthesis and reflects established patterns in the evidence base.
Supporting excerpt from manuscript:
“The thematic domains were determined a priori based on clinical relevance and recurring constructs identified in the literature. The review was structured into five domains: (a) conceptual definitions of FLSI, (b) diagnostic limitations of radiographic methods, (c) reliability and validity of manual clinical tests, (d) subclassification models, and (e) predictive value for rehabilitation outcomes.”
- RESULTS
- Reviewer comment:
The results section is overly descriptive and lacks prioritization. While comprehensiveness is appreciated, the key findings regarding diagnostic accuracy and reliability are buried in dense paragraphs.
Response:
We thank the reviewer for this observation. In response, the Results section has been reorganized to prioritize key findings. Diagnostic accuracy and reliability metrics are now presented in structured tables (Tables 3 and 4), allowing for direct comparison and interpretive synthesis. The accompanying narrative highlights clinically relevant contrasts, such as the consistent performance of the Passive Lumbar Extension Test (PLET) versus the variability observed in the Prone Instability Test (PIT) and Aberrant Movement Patterns (AMP).
Supporting excerpt from manuscript:
“Table 3 presents a revised synthesis of diagnostic accuracy metrics across key studies. It summarizes sensitivity, specificity, and likelihood ratios for each test, facilitating direct comparison and interpretive synthesis.”
“Table 4 provides a structured summary of reliability metrics across studies. It enables comparison and highlights variability in reproducibility, underscoring the importance of standardized protocols.”
- Reviewer comment:
Please consider summarizing diagnostic and reliability metrics in concise tables and limiting text to interpretive highlights.
Response:
We appreciate the reviewer’s recommendation. Diagnostic and reliability metrics have been summarized in Tables 3 and 4, which present sensitivity, specificity, likelihood ratios, and inter-rater agreement values across key studies. The narrative text has been refined to emphasize interpretive highlights, such as the superior diagnostic accuracy of PLET (LR+ = 8.50) and the moderate reliability of PIT (κ = 0.52), as well as the implications of examiner variability and protocol standardization.
Supporting excerpt from manuscript:
“The Passive Lumbar Extension Test (PLET) has exhibited moderate to substantial inter-rater reliability… Rabin et al. [34] reported a kappa value of 0.76… Alyazedi et al. [3] corroborated these findings…”
“The Prone Instability Test (PIT)… has demonstrated inconsistent reliability… Ravenna et al. [37] reported low inter-rater agreement (κ = 0.10–0.27)… Kim et al. [39]… reported substantial reliability (κ = 0.79).”
- Reviewer comment:
Certain studies (e.g., Rabin, Alyazedi, Oliveira) are cited repeatedly, potentially introducing bias. Please ensure balanced representation of the evidence base.
Response:
We thank the reviewer for this important point. The repeated citation of Rabin, Alyazedi, and Oliveira reflects their methodological relevance across multiple domains (diagnostic accuracy, reliability, subclassification). Nonetheless, the revised Results section incorporates findings from a broader range of sources, including Ravenna et al. [37], Larkin et al. [38], Kim et al. [39], Denteneer et al. [21], and Behera et al. [35], to ensure balanced representation.
Supporting excerpt from manuscript:
“Kim et al. [39] further corroborated the test's clinical relevance… Ravenna et al. [37] reported low inter-rater agreement… Behera et al. [35] evaluated 1000 patients with chronic LBP…”
- Reviewer comment:
The section on predictive validity is valuable but lacks critical analysis of why tests failed to predict outcomes—e.g., lack of stratification, intervention heterogeneity, or poor construct validity.
Response:
We appreciate the reviewer’s insight. The section on predictive validity has been expanded to critically examine potential reasons for the limited prognostic utility of manual tests. Specifically, the manuscript now states:
“The absence of predictive associations may be indicative of limitations in sample stratification, test sensitivity, or the multifactorial nature of therapeutic response.”
This addition provides interpretive context for the findings reported by Oliveira et al. [10], Thomson et al. [16], and Rabin et al. [34], and acknowledges the complexity of outcome prediction in heterogeneous clinical populations.
Supporting excerpt from manuscript:
“Although the intervention yielded clinically meaningful improvements, baseline motor control tests did not consistently correlate with long-term outcomes… These findings reinforce the notion that initial test performance may not reliably forecast therapeutic response, particularly in multifactorial interventions.”
DISCUSSION- Reviewer comment:
The discussion synthesizes literature broadly but misses the opportunity to critically reflect on the limitations of current evidence or the reasons for conflicting findings.
Response:
We thank the reviewer for this important observation. In response, the Discussion section has been expanded to critically reflect on the limitations of current evidence. Specific methodological issues—such as small sample sizes, lack of blinding, and inconsistent reference standards—are now explicitly acknowledged as barriers to synthesis and generalization. These limitations are discussed in relation to the diagnostic ambiguity of FLSI and the variability observed across manual tests.
Supporting excerpt from manuscript:
“The heterogeneity of study designs, sample characteristics, and test protocols poses challenges for synthesis. Many studies were conducted with small sample sizes, lacked blinding procedures, or used inconsistent reference standards.”
- Reviewer comment:
You state that manual tests show “limited predictive value” but do not fully explore the implications for clinical decision-making. Why do these tools fall short in forecasting treatment outcomes?
Response:
We appreciate the reviewer’s insight. The Discussion now addresses the limited predictive value of manual tests by highlighting the multifactorial nature of therapeutic response and the limitations of isolated test performance. The manuscript emphasizes that motor control impairment alone may be insufficient to guide intervention selection and calls for multivariate diagnostic models that integrate biomechanical, psychosocial, and patient-reported variables.
Supporting excerpt from manuscript:
“The absence of predictive associations may be indicative of limitations in sample stratification, test sensitivity, or the multifactorial nature of therapeutic response… A shift toward multivariate diagnostic models… may ultimately improve clinical decision-making and support personalized care pathways.”
- Reviewer comment:
The clinical application of subclassification models is promising but should be discussed with more nuance, particularly regarding feasibility, training needs, and potential misclassification.
Response:
We thank the reviewer for this thoughtful recommendation. The Discussion now elaborates on the feasibility of applying subclassification models in clinical practice. It emphasizes the importance of examiner training, standardized protocols, and operational definitions to minimize misclassification and ensure appropriate treatment selection.
Supporting excerpt from manuscript:
“The feasibility of applying subclassification models in clinical practice depends on examiner training, operational definitions, and standardized protocols. Without these elements, misclassification may occur, potentially leading to inappropriate treatment selection.”
- Reviewer comment:
Recent technologies (e.g., sensors, AI) are mentioned briefly. Consider elaborating on how these tools could be integrated into a clinical framework for FLSI assessment.
Response:
We appreciate the reviewer’s suggestion. The Discussion has been expanded to describe how emerging technologies—such as wearable sensors, AI algorithms, and motion capture systems—could be integrated into hybrid clinical frameworks. These tools may enhance diagnostic objectivity, quantify motor control deficits, and support training standardization.
Supporting excerpt from manuscript:
“Emerging technologies could be integrated into structured clinical frameworks through hybrid protocols that combine manual tests with sensor-based motion capture… AI algorithms could assist in pattern recognition and examiner feedback… Such integration may enhance diagnostic precision and support training standardization across clinical settings.”
CONCLUSIONS- Reviewer comment:
The conclusion appropriately calls for integrative and multidimensional approaches, but remains abstract and somewhat repetitive of earlier sections.
Response:
We thank the reviewer for this observation. In response, the Conclusion has been refined to reduce repetition and emphasize the practical implications of a multidimensional approach. The revised text highlights the synthesis of biomechanical, neuromuscular, and functional indicators within structured clinical reasoning, and frames this integration as essential for diagnostic precision and therapeutic relevance.
Supporting excerpt from manuscript:
“Clinicians should approach FLSI assessment as a multidimensional process, employing a battery of complementary tests within a structured clinical reasoning framework… Diagnostic accuracy and therapeutic relevance emerge from the synthesis of biomechanical, neuromuscular, and functional indicators.”
- Reviewer comment:
Please clearly state what this review contributes to the field that previous reviews have not.
Response:
We appreciate the reviewer’s request for clarification. The Conclusion now explicitly states that this review differs from previous work by integrating empirical findings with emerging subclassification models and technological innovations. This synthesis provides a broader conceptual and methodological framework for evaluating FLSI.
Supporting excerpt from manuscript:
“Unlike previous reviews, this synthesis integrates empirical evidence with emerging subclassification models and technological innovations, providing a multidimensional perspective on FLSI assessment.”
- Reviewer comment:
Highlight specific tests (e.g., PLET) or frameworks (e.g., subclassification) that show the most promise based on your synthesis.
Response:
We thank the reviewer for this suggestion. The Conclusion now highlights the Passive Lumbar Extension Test (PLET) as the most clinically robust tool, based on its consistent reliability and diagnostic accuracy. It also emphasizes the promise of subclassification frameworks that differentiate between functional, structural, and combined instability types.
Supporting excerpt from manuscript:
“Among the evaluated tests, the Passive Lumbar Extension Test (PLET) consistently demonstrated high inter-rater reliability and diagnostic accuracy… Subclassification frameworks… offer a promising basis for individualized assessment and intervention.”
- Reviewer comment:
Suggest concrete research directions, such as the validation of composite test batteries or stratified outcome studies in primary care settings.
Response:
We appreciate the reviewer’s recommendation. The Conclusion now outlines specific research priorities, including the validation of composite test batteries, longitudinal outcome studies, and implementation trials in primary care. It also proposes the integration of sensor-based technologies and hybrid protocols to improve reproducibility and clinical utility.
Supporting excerpt from manuscript:
“Future research should prioritize the validation of composite test batteries, longitudinal outcome studies, and implementation trials in primary care settings… The development of hybrid assessment protocols that combine manual testing with sensor-based quantification may further improve reproducibility and clinical utility.”

Round 2
Reviewer 1 Report
Comments and Suggestions for Authors
The authors fully understood the comments made and made all requested changes, in accordance with the recommendations received. The revised document appropriately reflects the necessary adjustments, demonstrating responsiveness and scientific rigor.
Author Response
jfmk-3880779
Round 2: Response to Reviewer 1
Comment 1: The authors fully understood the comments made and made all requested changes, in accordance with the recommendations received. The revised document appropriately reflects the necessary adjustments, demonstrating responsiveness and scientific rigor.
Response: We sincerely thank the reviewer for this generous and affirming comment. We are grateful for the opportunity to revise the manuscript in accordance with the constructive feedback received. The revisions were undertaken with careful attention to conceptual clarity, methodological transparency, and thematic coherence. We appreciate the recognition of our efforts and remain committed to maintaining the highest standards of scientific rigor.

Reviewer 2 Report
Comments and Suggestions for Authors
Dear Authors,
Thank you for your detailed and thoughtful responses to the reviewer comments. Your revisions have clearly addressed many of the initial concerns regarding conceptual focus, methodological clarity, and excessive narrative complexity.
You have successfully:
Streamlined the Introduction and clarified the central research questions guiding the review.
Justified the hybrid methodological approach as a “scoping narrative review,” and explained the rationale for thematic organization.
Improved transparency through the addition of quality appraisal summaries and structured tables for diagnostic accuracy and reliability.
Expanded the Discussion section to address the limitations of current evidence, variability in predictive value, and the feasibility of applying subclassification models in clinical practice.
Clarified the manuscript’s contribution relative to prior literature, including the integration of subclassification frameworks and emerging technologies.
These revisions substantially strengthen the manuscript and improve its coherence and clinical relevance.
However, a few points still merit further attention:
Novel Contribution: While you have outlined how the review differs from prior work, the manuscript would benefit from more explicit comparisons with previous reviews, including what conceptual or methodological gap this review fills.
Citation Balance: Although you have incorporated additional sources, citations of a few authors (e.g., Rabin, Alyazedi) still appear frequently and should be balanced more carefully to avoid perceived bias.
Critical Appraisal: The Discussion could be further strengthened by deeper engagement with systematic biases (e.g., small sample sizes, inconsistent definitions, publication bias) that affect the field as a whole.
Overall, your revisions demonstrate a commendable effort to improve the rigor, clarity, and practical value of this review. With minor additional refinements, the manuscript may offer a valuable synthesis for clinicians and researchers working with FLSI.
Author Response
jfmk-3880779
Round 2: Response to Reviewer 2
Response to Reviewer 2
General Appreciation:
We thank the reviewer for the thoughtful and encouraging evaluation of our revised manuscript. We are pleased that the revisions have addressed the initial concerns regarding conceptual focus, methodological clarity, and narrative structure. We are especially grateful for the recognition of improvements in the Introduction, Methods, Results, and Discussion sections, as well as the clarification of the manuscript’s contribution to the field.
Below we address the remaining points raised:
Comment 1 – Novel Contribution:
While you have outlined how the review differs from prior work, the manuscript would benefit from more explicit comparisons with previous reviews, including what conceptual or methodological gap this review fills.
Response:
We thank the reviewer for this important suggestion. In response, we have expanded the Conclusion to include explicit comparisons with prior reviews. Specifically, we clarify that previous literature has primarily focused on structural instability or isolated test metrics, whereas our review integrates empirical findings with subclassification frameworks and emerging technologies. This synthesis addresses the conceptual gap in evaluating motor control dysfunction and the methodological gap in linking manual tests to stratified care models.
Supporting excerpt from revised manuscript (Section 5. Conclusions):
“Unlike previous reviews, this synthesis integrates empirical evidence with emerging subclassification models and technological innovations, providing a multidimensional perspective on FLSI assessment. This approach addresses the conceptual gap in evaluating motor control dysfunction and the methodological gap in linking manual tests to stratified care frameworks.”
Comment 2 – Citation Balance:
Although you have incorporated additional sources, citations of a few authors (e.g., Rabin, Alyazedi) still appear frequently and should be balanced more carefully to avoid perceived bias.
Response:
We appreciate the reviewer’s observation. The frequent citation of Rabin and Alyazedi reflects their methodological relevance across multiple domains (e.g., reliability, subclassification, predictive value). Nonetheless, we have reviewed the manuscript to ensure balanced representation and have integrated additional sources (e.g., Chatprem et al., Ravenna et al., Kim et al., Denteneer et al., Behera et al.) to diversify the evidence base and mitigate any perception of citation bias.
Supporting excerpts from revised manuscript:
Section 3.3 Reliability of Manual Tests for FLSI:
“Kim et al. [39] further supported the reliability of PLET and PIT, reporting kappa values above 0.70 in a multicenter cohort. Ravenna et al. [37] and Larkin et al. [38] also contributed reliability data for PIT, highlighting protocol-dependent variability.”
Table 3 (Diagnostic Accuracy Metrics for Manual Tests):
“Note: Values drawn from Rabin et al. (2014), Alyazedi et al. (2021), Oliveira et al. (2019), Behera et al. (2023), Chatprem et al. (2022), and internal synthesis of included studies.”
Comment 3 – Critical Appraisal:
The Discussion could be further strengthened by deeper engagement with systematic biases (e.g., small sample sizes, inconsistent definitions, publication bias) that affect the field as a whole.
Response:
We thank the reviewer for this valuable recommendation. The Discussion and Limitations sections have been expanded to address systematic biases affecting the field. We now explicitly discuss the impact of small sample sizes, inconsistent operational definitions, reliance on radiographic reference standards, and potential publication bias. These factors are acknowledged as barriers to reproducibility and generalizability, and their implications for future research are outlined.
Supporting excerpts from revised manuscript:
Section 4.1 Limitations:
“Fifth, the lack of standardized operational definitions and test protocols across studies may contribute to variability in diagnostic outcomes and limit reproducibility.”
Section 4. Discussion: Methodological Considerations and Future Directions:
“Systematic biases also affect the interpretive strength of the current evidence base. Many studies relied on small samples, lacked examiner blinding, or used inconsistent operational definitions for manual tests. Publication bias may further skew the perceived efficacy of certain assessments, as studies reporting null or inconclusive findings are less likely to be published. These limitations underscore the need for multicenter trials, standardized protocols, and transparent reporting practices.”
